# Factors Affecting the Necessity of Tracheostomy in Patients with Deep Neck Infection

**DOI:** 10.3390/diagnostics11091536

**Published:** 2021-08-25

**Authors:** Shih-Lung Chen, Chi-Kuang Young, Tsung-You Tsai, Huei-Tzu Chien, Chung-Jan Kang, Chun-Ta Liao, Shiang-Fu Huang

**Affiliations:** 1Department of Otorhinolaryngology & Head and Neck Surgery, Chang Gung Memorial Hospital, Linkou 333, Taiwan; rlong289@gmail.com (S.-L.C.); mp0594@cgmh.org.tw (T.-Y.T.); handneck@gmail.com (C.-J.K.); liaoct@cgmh.org.tw (C.-T.L.); 2School of Medicine, Chang Gung University, Taoyuan 333, Taiwan; rioriorioman@gmail.com (C.-K.Y.); kathy.htchien@gmail.com (H.-T.C.); 3Department of Otorhinolaryngology, Chang Gung Memorial Hospital, Keelung 204, Taiwan; 4Department of Nutrition and Health Sciences, Chang Gung University of Science and Technology, Taoyuan 333, Taiwan; 5Graduate Institute of Clinical Medical Sciences, Chang Gung University, Taoyuan 333, Taiwan

**Keywords:** deep neck infection, tracheostomy, parapharyngeal space, retropharyngeal space, *Streptococcus constellatus*

## Abstract

Deep neck infection (DNI) is a serious disease that can lead to airway obstruction, and some patients require a tracheostomy to protect the airway instead of intubation. However, no previous study has explored risk factors associated with the need for a tracheostomy in patients with DNI. This article investigates the risk factors for the need for tracheostomy in patients with DNI. Between September 2016 and February 2020, 403 subjects with DNI were enrolled. Clinical findings and critical deep neck spaces associated with a need for tracheostomy in patients with DNI were assessed. In univariate and multivariate analysis, older age (≥65 years old) (OR = 2.450, 95% CI: 1.163–5.161, *p* = 0.018), multiple spaces involved (≥3 spaces) (OR = 4.490, 95% CI: 2.153–9.360, *p* = 0.001), and the presence of mediastinitis (OR = 14.800, 95% CI: 5.097–42.972, *p* < 0.001) were independent risk factors associated with tracheostomy in patients with DNI. Among the 44 patients with DNI that required tracheostomy, ≥50% of patients had involvement of the parapharyngeal or retropharyngeal space (72.72% and 50.00%, respectively). *Streptococcus constellatus* (25.00%) was the most common pathogen in patients with DNI who required tracheostomy. In conclusion, requiring a tracheostomy was associated with a severe clinical presentation of DNI. Older age (≥65 years old), multiple spaces (≥3 spaces), and presence of mediastinitis were significant risk factors associated with tracheostomy in patients with DNI. The parapharyngeal and retropharyngeal spaces were the most commonly involved, and *Streptococcus constellatus* was the most common pathogen in the patients with DNI that required tracheostomy.

## 1. Introduction

Deep neck infection (DNI) is a fatal bacterial infection in the potential spaces of the neck [1]. DNI can lead to airway obstruction and cause severe morbidity and mortality. Securing the airway is critical to manage DNI [2]. Intubation and tracheostomy are the most common types of airway management strategies. In clinical practice, tracheostomy is indicated when conventional intubation is difficult, such as with obvious signs of an obstructed upper airway, severe oral trismus, distorted anatomy of the neck, serious pharyngeal wall bulging, or critical laryngeal edema. In addition, prolonged intubation of more than 2 weeks may be an indication for tracheostomy.

However, sometimes, decisions of tracheotomy and intubation are based on the physician’s judgment. Currently, there are no clear objective factors to help physicians determine when to perform tracheostomy, especially when the sign of airway obstruction is not obvious. Thus, the purpose of this research was to analyze the risk factors associated with performance of tracheostomy in patients with DNI.

## 2. Materials and Methods

This study retrospectively reviewed the medical records of 403 patients diagnosed with DNI who were admitted to Chang Gung Memorial Hospital in Linkou, Taiwan, between September 2016 and February 2020. Diagnostic imaging procedures included ultrasonography (US) and computed tomography (CT). The treatment course included antibiotics, US-guided needle drainage, and open surgical incision and drainage. The empirical antibiotics used were ceftriaxon 1 gm Q12h and metronidazole 500 mg Q8h according to previous reports to cover aerobic and anaerobic bacteria before the culture results were available [3,4]. In this research, six physicians were responsible for the treatment of patients with DNI. Each patient had a physician responsible for surgical treatment. Each physician independently decided whether the patient should undergo tracheostomy.

To investigate the risk factors associated with need for a tracheostomy, we collected the patient’s gender, age, chief complaint period, hospital-staying period, C-reactive protein (CRP), blood sugar, diabetes mellitus (DM) status, performance of incision and drainage surgery, results of US-guided drainage, number of spaces affected by DNI, presence of mediastinitis, and performance of tracheostomy or not. 

Based on Centers for Disease Control and Prevention guidelines [5], the definition of mediastinitis needs one of the following criteria: organisms cultured from mediastinal tissue or fluid, evidence of mediastinitis on gross anatomical or histopathological examination, or one of the following signs or symptoms: chest pain, fever (>38 °C), or sternal instability. In addition, it requires at least one of the following: mediastinal widening on imaging or purulent drainage from mediastinal area [6].

### 2.1. Exclusion Criteria

Patients with cervical necrotizing fasciitis, severe cardiopulmonary diseases, limited intraoral abscesses, and peri-tonsillar abscesses and those who were immunocompromised were excluded. A total of 403 patients were enrolled.

### 2.2. Statistical Analysis

All data were analyzed using MedCalc software (ver. 18.6; MedCalc, Ostend, Belgium). The Kolmogorov–Smirnov test showed that the data were not normally distributed; thus, we employed the chi-square test for categorical variables, and logistic regression was performed for the univariate and multivariate analysis. A multivariate forward stepwise selection procedure was implemented, and all of the variables included in univariate analysis were entered into the final multivariate model. A *p* value < 0.05 was considered to reflect statistical significance.

## 3. Results

Demographic and clinical data are shown in Table 1. A total of 403 patients with DNI were enrolled from September 2016 to February 2020. A total of 271 males (67.24%) and 132 females (32.76%) had a mean age of 51.29 ± 18.92. The mean chief complaint and hospital-staying period were 5.04 ± 4.53 and 9.66 ± 7.98 days, respectively. For the laboratory data, mean CRP was 140.15 ± 107.72 (mg/L), and mean blood sugar was 142.81 ± 72.89 (mg/dL). A total of 156 (38.70%) patients had DM.

For treatment procedures, 187 patients underwent an incisional and drainage surgery for DNI (46.40%), and 71 people underwent an US-guided drainage procedure for DNI (17.61%). Among these patients, 190 (47.14%) patients had single spaces involved, 98 (24.33%) had double spaces involved, and 115 (28.53%) had more than three spaces involved. Mediastinitis was found in 21 (5.21%) patients. Tracheostomy was performed in 44 (10.91%) patients. In Table 2, we show the univariate analyses of variables for 403 patients and found that older age (≥65 years old), higher CRP level, multiple spaces (≥3 spaces), and presence of mediastinitis were significantly associated with a need for tracheostomy (*p* < 0.05). 

There were 96 patients in the ≥65 years old group, and 19 underwent tracheostomy. However, there were 307 patients in the <65 years group, and only 25 required a tracheostomy. This difference was significant (OR = 2.783, 95% CI: 1.456–5.318, *p* = 0.001). Furthermore, patients that underwent tracheostomy had an average mean CRP of 180.15 ± 103.46 mg/L. However, CRP for those without tracheostomy was 135.24 ± 103.46 mg/L (OR = 1.003, 95% CI: 1.000–1.006, *p* = 0.011). In addition, multiple space involvement (≥3 spaces) was a significant risk factor for requiring a tracheostomy (OR = 6.907, 95% CI: 3.501–13.62, *p* < 0.001). The presence of mediastinitis was another significant risk factor for requiring a tracheostomy (OR = 23.46, 95% CI: 8.799–62.58, *p* < 0.001).

In Table 2, all factors were entered into a forward stepwise multivariate logistic regression model. Older age (≥65 years old) (OR = 2.450, 95% CI: 1.163–5.161, *p* = 0.018), multiple spaces (≥3 spaces) (OR = 4.490, 95% CI: 2.153–9.360, *p* = 0.001), and presence of mediastinitis (OR = 14.800, 95% CI: 5.097–42.972, *p* < 0.001) were significant independent risk factors for tracheostomy in patients with DNI.

Table 3 shows a comparison of infected spaces and pathogens between 44 patients with tracheostomy and 359 patients without tracheostomy. There were no significant differences between these two groups. In the tracheostomy group, ≥50% patients showed involvement of the parapharyngeal space or retropharyngeal space (72.72% and 50.00%, respectively).

On the other hand, in 44 patients with tracheostomy, pathogens that accounted for ≥5% of culture results included (listed sequentially): *Streptococcus constellatus* (25.00%), *Streptococcus anginosus* (20.45%), *Prevotella buccae* (15.90%)*, Klebsiella pneumonia* (13.63%), *Prevotella intermedia* (9.09%), *Parvimonas micra* (6.81%), and *Staphylococcus aureus* (6.81%). Among these patients, 7 (15.90%) had no growth of specific pathogens.

## 4. Discussion

DNI is an important infectious disease with high morbidity and mortality and can lead to severe life-threatening complications, such as airway obstruction, pneumonia, lung abscess, sepsis, pericarditis, internal jugular vein thrombosis, and carotid artery erosion [4,7]. The mortality rate even reached 40% to 50% if such complications occurred [8]. 

As shown in Table 1, males accounted for 67.24% and females accounted for 32.76% of DNIs; this male predominance was also observed in previous studies [3,9]. The average age of our patients was 51.29 ± 18.92 years, similar to previous studies [4,9]. Older patients have more comorbidities, resulting in more complicated clinical situations. Chang et al. reported a DNI-related mortality of 1–2.5%, and the mortality rate in patients with comorbidities or old age tended to be higher (1.5–5%) [10]. Yang et al. concluded that the hospital stay was also longer in patients older than 65 years, at which point antibiotic treatment may be inadequate or ineffective for DNI [11]. Clinical conditions were more complicated in older patients than in younger patients, and the chances of undergoing tracheostomy to secure the airway was higher, suggesting that age is a risk factor.

CRP, which is synthesized by hepatocytes, is an acute inflammatory protein released during infectious processes in response to pro-inflammatory cytokines [12]. Wang et al. reported that patients with DNI and a CRP of more than 100 mg/L have longer hospital stays [8]. As shown in Table 2, patients that underwent tracheostomy had an average mean CRP of 180.15 ± 103.46 mg/L, about 1.5-fold greater than patients who did not undergo tracheostomy (CRP level: 135.24 ± 103.46 mg/L, *p* = 0.011). However, CRP did not reach statistical significance in multivariate analysis. Indeed, the higher level of CRP was representative of more severe infection. However, it did not necessarily mean that DNI would lead to airway obstruction. If severe abscess did not occur at a critical site and result in a compromised airway, tracheostomy was not required (even with a high CRP level).

Our results indicate that DNI involving multiple spaces (≥3 spaces) is an independent risk factor for the need for tracheostomy. Velhonoja et al. reported that infection would be severely advanced when the DNI involved multiple spaces [1]. In Maharaj et al., multiple deep neck space involvement was a risk factor for patient mortality with DNI, and the elderly patients with DNIs tended to have more multiple space involvement than the patients under 18 years old [13]. In fact, multiple-space DNI puts the airway at risk and increases the chance of airway compromise. Therefore, when treating the patients with DNI involving multiple spaces, tracheostomy is a common strategy to protect the airway.

In previous studies, the rate of mediastinitis with DNI was reported to be 1.7–5.4% [10,14]. In our study, 21 (5.21%) of 403 patients developed mediastinitis (Table 1), but 14 (31.81%) of 44 patients that developed mediastinitis underwent tracheostomy (Table 2). The presence of mediastinitis was significant in both univariate and multivariate analysis as a risk factor for the need for tracheostomy. When the mediastinum was severely infected with micro-abscesses and gas formation, the airway was compromised and obstruction becomes possible (Figure 1). In addition to airway compromise, infective mediastinal extension is quite commonly associated with higher morbidity, with a mortality rate of around 40% [15]. A patient with mediastinitis may have a higher rate of tracheostomy due to prolonged admission and prolonged airway-securing. Indeed, Tapiovaara et al. showed that extension of the infection to the mediastinal space increased the length of tracheotomy dependence and the total length of hospital care [2]. 

In DNI, DM was a common underlying disease and was related to poor outcomes [4,16,17,18,19]. In Table 1, the average blood sugar level was 142.81 ± 72.89, and about 38.70% of 403 patients had DM. Although the blood sugar level (*p* = 0.217) and presence of DM (*p* = 0.107) did not achieve the statistical significance in UVA for the need for tracheostomy (Table 2), we found that patients who underwent tracheostomy had relatively high blood sugar (156.31 ± 83.52) compared with those who did not require tracheostomy (141.16 ± 71.44). Therefore, control of blood sugar and management of DM is important for DNI treatment.

The parapharyngeal space and retropharyngeal space (Figure 2) were the most common spaces involved that resulted in tracheostomy (Table 3). The deep neck spaces lie within a complex framework formed by the cervical fascial planes [20]. The parapharyngeal space is shaped like an inverted pyramid and communicates with other cervical and cranial fascial spaces, as well as with the mediastinum [21]. On the other hand, the retropharyngeal space is a potential space of the head and neck, bound by the buccopharyngeal fascia anteriorly and the alar fascia posteriorly. Serious infections of the teeth can spread down this space into the posterior mediastinum [22]. In fact, the parapharyngeal space is next to the trachea, and the retropharyngeal space is just behind the trachea. With these two spaces involved, the trachea may become distorted and compressed, which can further hinder conventional laryngoscopy and intubation. Garcia et al. even mentioned that the retropharyngeal and the parapharyngeal spaces put a patient at high risk for complications. This is probably due to infections from these two spaces through the fascia, which could lead to mediastinal infection [23]. Therefore, parapharyngeal space and retropharyngeal space involvement are more likely to cause airway compromise than other spaces.

DNI have various clinical presentations depending on the pathogenic organism [24]. In Table 3, *Streptococcus constellatus* (25%) was the most cultivated pathogen in the 44 patients who underwent tracheostomy. This pathogen belongs to the Streptococcus milleri group, a subgroup of viridans streptococci that consists of three streptococcal species—namely, Streptococcus anginosus, Streptococcus intermedius, and *Streptococcus constellatus* [24,25,26]. 

These organisms are commonly found on the mucous membrane of the oral cavity, oral pharynx, gastrointestinal tract, and genitourinary tract. Although they are commensal organisms, they can become pathogenic after mucosal disruption, form abscesses, and have local aggressive extension to surrounding tissues such as deep neck spaces, or even be associated with severe suppurative infection [27]. Han et al. reported that surgical drainage with antibiotics is generally successful for the management of these conditions. However, there has been decreasing susceptibility of *Streptococcus milleri* group to penicillin, cephalosporins, macrolides, ciprofloxacin, and clindamycin [24]. For clinical practice, effective broad-spectrum antibiotics against causative organisms are still important in managing DNI [16]. Although pus cultures obtained from a needle aspiration or surgical drainage can guide antibiotic choice, broad-spectrum empirical antibiotics are important before culture results are available. In this research, there were still 15.90% of patients who did not show definite growth of pathogens. The negative culture results may be due to the use of antimicrobial agents prior to admission or intravenous antibiotic treatment before surgical treatment [3].

### Limitations of the Article

There were some limitations to our retrospective study. There was no internal guideline available during the study period to guide the indication for tracheostomy in our hospital. Our physicians made a comprehensive judgment based on the patient’s vital signs, respiratory condition, blood oxygen saturation, laboratory data, imaging examination, and the severity of DNI to decide whether to perform tracheostomy or intubation to protect the airway. The decision for tracheostomy is clinical and determined by the surgeon. As such, while the independent variables are logical for tracheostomy placement, they do not mandate the requirement for tracheotomy. The selection bias occurred inherently in this mode of data acquisition. Prospective, randomized controlled studies are required to support our results and address these limitations.

## 5. Conclusions

Requiring a tracheostomy in DNI was associated with severe clinical presentations. Older age (≥65 years old), multiple spaces (≥3 spaces), and presence of mediastinitis were significant risk factors associated with the need for tracheostomy in patients with DNI in both univariate and multiple analysis. The parapharyngeal and retropharyngeal spaces were the most common two spaces involved. *Streptococcus constellatus* was the most common pathogen in patients requiring tracheostomy for DNI.

## Figures and Tables

**Figure 1 diagnostics-11-01536-f001:**
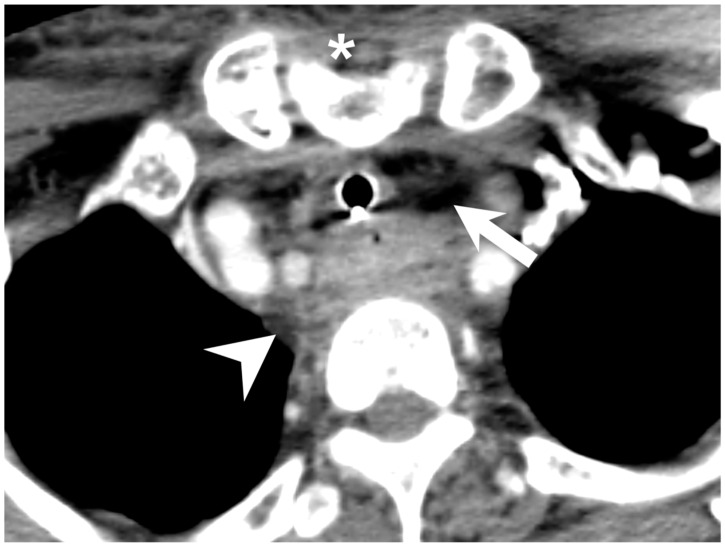
A patient with deep neck infection and mediastinitis with conduction of tracheostomy; micro-abscess (arrowhead) and gas (arrow) formation was noted. Sternum notch (asterisk).

**Figure 2 diagnostics-11-01536-f002:**
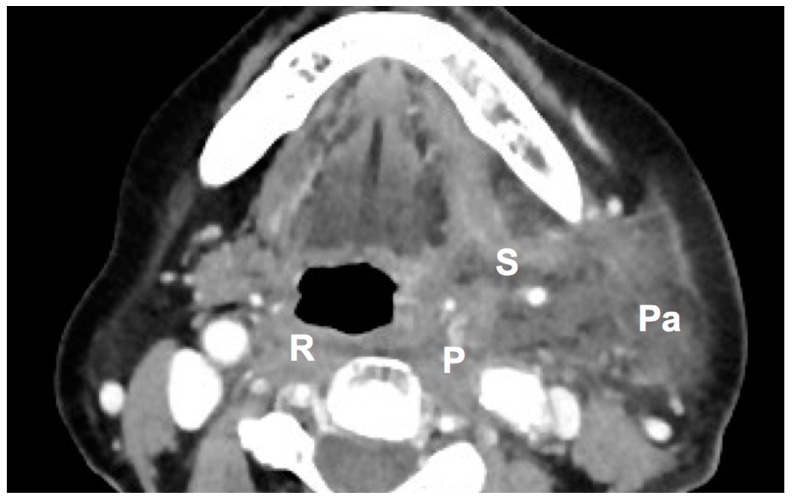
The axial view of CT from a patient with multiple deep neck spaces involved. There were abscess formations in retropharyngeal space, parapharyngneal space, submandibular space, and parotid space. (R: retropharyngeal space; P: parapharyngeal space; S: submandibular space; Pa: parotid space).

**Table 1 diagnostics-11-01536-t001:** Clinicopathological characteristics of the 403 patients with deep neck infection.

Characteristics	N (%)
Gender	403 (100.00)
Male	271 (67.24)
Female	132 (32.76)
Age, years (SD)	51.30 ± 18.93
Chief complaint period, days (SD)	5.04 ± 4.53
Hospital-staying period, days (SD)	9.66 ± 7.98
CRP, mg/L (SD)	140.15 ± 107.72
Blood sugar, mg/dL (SD)	142.81 ± 72.89
Diabetes mellitus	156 (38.70)
Incision and drainage open surgery	187 (46.40)
Ultrasonography-guided drainage	71 (17.61)
Single space	190 (47.14)
Double spaces	198 (24.33)
Multiple spaces, ≥3	115 (28.53)
Mediastinitis	21 (5.21)
Tracheostomy performance	44 (10.91)

N = number; SD = standard deviation; CRP = C-reactive protein (normal range < 5 mg/L); blood sugar (normal range: 70–100 mg/dL).

**Table 2 diagnostics-11-01536-t002:** Univariate and multivariate analysis of tracheostomy performance in 403 patients with deep neck infection.

Variable	Tracheostomy Performance		Univariate Analysis			Multivariate Analysis		
	Yes	No	OR	95% CI	*p* Value	OR	95% CI	*p* Value
Gender	44	359			0.405	-	-	-
Male	32	239	1.000					
Female	12	120	0.476	0.371–1.502				
Age, years					0.001 *			0.018 *
<65	25	282	1.000					
≥65	19	77	2.783	1.456–5.318		2.450	1.163–5.161	
Chief complaint days (SD)	4.75 ± 103.32	5.08 ± 104.66	0.982	0.910–1.059	0.634	-	-	-
CRP, mg/L (SD)	180.15 ± 103.46	135.24 ± 103.46	1.003	1.000–1.006	0.011 *	-	-	-
Blood sugar, mg/dL (SD)	156.31 ± 183.52	141.16 ± 171.44	1.002	0.998–1.006	0.217	-	-	-
Diabetes mellitus					0.107	-	-	-
No	22	225	1.000					
Yes	22	134	1.679	0.895–3.147				
Multiple spaces, ≥3					<0.001 *			0.001 *
No	14	274	1.000					
Yes	30	85	6.907	3.501–13.62		4.490	2.153–9.360	
Mediastinitis					<0.001 *			<0.001 *
No	30	352	1.000					
Yes	14	7	23.46	8.799–62.58		14.800	5.097–42.972	

SD = standard deviation; OR = odds ratio; CI = confidence intervals; CRP = C-reactive protein; * *p* < 0.05.

**Table 3 diagnostics-11-01536-t003:** Comparison of involved spaces and pathogens between 44 patients with tracheostomy and 359 patients without tracheostomy in deep neck infection.

Characteristics	Tracheostomy, N (%)	Non-Tracheostomy, N (%)	*p* Value
Location			
Parapharyngeal space	32 (72.72)	210 (58.49)	0.074
Retropharyngeal space	22 (50.00)	132 (36.76)	0.100
Submandibular space	20 (45.45)	165 (45.96)	1.000
Masticator space	14 (31.81)	75 (20.89)	0.122
Anterior cervical space	9 (20.45)	45 (12.53)	0.159
Parotid space	5 (11.36)	77 (21.44)	0.163
Perivertebral space	5 (11.36)	19 (5.29)	0.164
Carotid space	4 (9.09)	19 (5.29)	0.298
Visceral space	3 (6.81)	21 (5.84)	0.736
Posterior cervical space	1 (2.27)	7 (1.94)	1.000
Pathogens			
* Streptococcus constellatus*	11 (25.00)	93 (25.90)	1.000
* Streptococcus anginosus*	9 (20.45)	60 (16.71)	0.527
* Prevotella buccae*	7 (15.90)	43 (11.97)	0.467
* Klebsiella pneumonia*	6 (13.63)	50 (13.92)	1.000
* Prevotella intermedia*	4 (9.09)	25 (26.96)	0.541
* Parvimonas micra*	3 (6.81)	34 (9.47)	0.783
* Staphylococcus aureus*	3 (6.81)	14 (3.89)	0.414
* Pseudomonas aeruginosa*	1 (2.27)	7 (1.94)	1.000
* Staphylococcus epidemidis*	1 (2.27)	8 (2.22)	1.000
* Eikenella corrodens*	1 (2.27)	7 (1.94)	1.000
* Streptococcus salivarius*	1 (2.27)	8 (2.22)	1.000
* Slackia exigua*	1 (2.27)	11 (23.06)	1.000
* Streptococcus oralis*	1 (2.27)	6 (1.67)	0.557
* Gemella morbillorum*	1 (2.27)	9 (2.50)	1.000
No growth	7 (15.90)	98 (27.50)	0.144

N = number.

## Data Availability

All data generated or analyzed during this study are included in this published article. The data are available on request.

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
