# Peer review of "Factors Affecting the Necessity of Tracheostomy in Patients with Deep Neck Infection"

_diagnostics, 2021, doi:10.3390/diagnostics11091536_

Round 1

Reviewer 1 Report

The manuscript presented to me for review concerns an important issue related to factors affecting the necessity of tracheostomy in patients with deep neck infection. I propose to complete the introduction in order to familiarize the reader with the analyzed problem in more detail.

Author Response

Revision of “Factors affecting the necessity of tracheostomy in patients with deep neck infection”

Dear Editor and reviewers,

Thank you for the thoughtful and detailed review. In the following pages, we respond in a point-by-point manner to all comments of the reviewers. In the revision, all changes are highlighted in RED color. During this valuable process of revision to enhance the manuscript, we really thank the editors and reviewers for their insightful and informative comments. Please let us know if additional clarification is required. We deeply appreciate your valuable comments, which encouraged us to consider our work more thoroughly. Thank you again.

Yours sincerely,

Corresponding author, Shiang-Fu Huang, MD, PhD

Department of Otorhinolaryngology & Head and Neck Surgery, Chang Gung Memorial Hospital, Linkou, Taiwan

Reviewer 1

The manuscript presented to me for review concerns an important issue related to factors affecting the necessity of tracheostomy in patients with deep neck infection.

Comment: I propose to complete the introduction in order to familiarize the reader with the analyzed problem in more detail.

Reply:

Yes, we appreciate the reviewer’s insightful comments. We completely agreed.

We revise the original last paragraph of the introduction from “However, no previous studies have explored risk factors for the need for tracheostomy in patients with DNI. Thus, we analyzed this topic.” to

“However, sometimes, decisions of tracheotomy and intubation are based on the physician’s judgment. Currently, there are no clear objective factors to help physicians determine when to perform tracheostomy, especially when the sign of airway obstruction is not obvious. Thus, the purpose of this article is to analyze the risk factors associated with performance of tracheostomy in patients with DNI.” on Page 1 last Lines 1-3 and Page 2 Lines 1-2.

Reviewer 2 Report

Good job

Author Response

Revision of “Factors affecting the necessity of tracheostomy in patients with deep neck infection”

Dear Editor and reviewers,

Thank you for the thoughtful and detailed review. In the following pages, we respond in a point-by-point manner to all comments of the reviewers. In the revision, all changes are highlighted in RED color. During this valuable process of revision to enhance the manuscript, we really thank the editors and reviewers for their insightful and informative comments. Please let us know if additional clarification is required. We deeply appreciate your valuable comments, which encouraged us to consider our work more thoroughly. Thank you again.

Yours sincerely,

Corresponding author, Shiang-Fu Huang, MD, PhD

Department of Otorhinolaryngology & Head and Neck Surgery, Chang Gung Memorial Hospital, Linkou, Taiwan

Reviewer 2

Comment: Good job

Reply:

Yes, we appreciate the reviewer’s insightful comments. We completely agreed.

We believe that this manuscript of “Factors affecting the necessity of tracheostomy in patients with deep neck infection” is appropriate for publication due to the meaningful and clinical significance.

Reviewer 3 Report

Single centre retrospective study (n=403) reporting risk factors necessitation tracheostomy in patients with deep neck infections. Well designed and presented study.

A few minor comments:

Add number of physicians involved into decision making for tracheostomy.

Add whether an internal guideline/SOP was available during study period to guide the indication for tracheostomy

Table 2 Sugar should be Blood sugar

Add how 'Mediastinitis' was defined

Page 6/8, lines 8-9 'retrospectivegneal space' is no known anatomic entity this reviewer is aware, please doublecheck and revise or comment

Page 7/8, Limitations: Please clarify 'resulted in a high attrition rate'

Correct few typos

Author Response

Revision of “Factors affecting the necessity of tracheostomy in patients with deep neck infection”

Dear Editor and reviewers,

Thank you for the thoughtful and detailed review. In the following pages, we respond in a point-by-point manner to all comments of the reviewers. In the revision, all changes are highlighted in RED color. During this valuable process of revision to enhance the manuscript, we really thank the editors and reviewers for their insightful and informative comments. Please let us know if additional clarification is required. We deeply appreciate your valuable comments, which encouraged us to consider our work more thoroughly. Thank you again.

Yours sincerely,

Corresponding author, Shiang-Fu Huang, MD, PhD

Department of Otorhinolaryngology & Head and Neck Surgery, Chang Gung Memorial Hospital, Linkou, Taiwan

Reviewer 3

Single centre retrospective study (n=403) reporting risk factors necessitation tracheostomy in patients with deep neck infections. Well designed and presented study.

A few minor comments:

Comment 1: Add number of physicians involved into decision making for tracheostomy.

Reply 1:

Yes, we appreciate the reviewer’s insightful comments. We completely agreed.

There are six clinical physicians in our author list, and each of them will decide whether to perform tracheostomy when facing the treatment of deep neck infection (DNI).

We added the relevant text to the Materials and Methods " In this research, six physicians were responsible for the treatment of patients with DNI. Each patient had a physician responsible for surgical treatment. Each physician will independently decide whether the patient should undergo tracheostomy surgery" on Page 2 Lines 11-13.

Comment 2: Add whether an internal guideline/SOP was available during study period to guide the indication for tracheostomy

Reply 2:

Yes, we thank for reviewer’ valuable comments.

We did not have an internal guideline/SOP available during study period to guide the indication for tracheostomy. Our physicians made a comprehensive judgment based on the patient’s vital sign, respiratory condition, blood oxygen saturation, laboratory data, imaging examination, and the severity of DNI to decide whether perform tracheostomy or intubation to protect the airway.

It’s the reason why we want to study whether there are objective independent factors that can help physicians judge whether to perform tracheostomy.

We added the relevant text to the limitation section “There was no internal guideline available during study period to guide the indication for tracheostomy in our hospital. Our physicians made a comprehensive judgment based on the patient’s vital sign, respiratory condition, blood oxygen saturation, laboratory data, imaging examination, and the severity of DNI to decide whether perform tracheostomy or intubation to protect the airway. The decision for tracheostomy is clinical, and determined by the surgeon.” on Page 7 Lines 17-22.

Comment 3: Table 2 Sugar should be Blood sugar.

Reply 3:

Yes, we thank for the reviewer’s comment. We totally appreciate.

We revise “sugar” of Table 2 to “blood sugar” (Page 3).

In addition, we also revise "sugar" of Table 1 (Page 2 and Page 3) and the whole article to "blood sugar". Thank reviewer for the detailed review.

Comment 4: Add how 'Mediastinitis' was defined

Reply 4:

Yes, we thank for reviewer’ comment. We totally agree.

We add the relevant text in the discussion section “Based on Centers for Disease Control and Prevention (CDC) guidelines [7], the definition of mediastinitis needs one of the following criteria: organisms cultured from mediastinal tissue or fluid, evidence of mediastinitis on gross anatomical or histopathological examination or one of the following signs or symptoms: chest pain, fever (>38°C) or sternal instability. In addition, it requires at least one of the following: mediastinal widening on imaging or purulent drainage from mediastinal area [8].” on Page 4 Lines 11-16 below Table 3.

New reference 7 and 8:

  1. Mangram, A.J.; Horan, T.C.; Pearson, M.L.; Silver, L.C.; Jarvis, W.R. Guideline for Prevention of Surgical Site Infection, 1999. Centers for Disease Control and Prevention (CDC) Hospital Infection Control Practices Advisory Committee. Am J Infect Control 1999, 27, 97-132; quiz 133-134; discussion 196.
  2. Abu-Omar, Y.; Kocher, G.J.; Bosco, P.; Barbero, C.; Waller, D.; Gudbjartsson, T.; Sousa-Uva, M.; Licht, P.B.; Dunning, J.; Schmid, R.A.; et al. European Association for Cardio-Thoracic Surgery expert consensus statement on the prevention and management of mediastinitis. Eur J Cardiothorac Surg 2017, 51, 10-29, doi:10.1093/ejcts/ezw326.

Comment 5: Page 6/8, lines 8-9 'retrospectivegneal space' is no known anatomic entity this reviewer is aware, please double check and revise or comment

Reply 5:

Yes, we appreciate the reviewer’s comments. We completely agreed.

'retrospectivegneal space' is our typo and we correct it to 'retropharyngeal space' on Page 6 Line 22. As the reviewer said, 'retrospectivegneal space' is no known anatomic entity.

Thank for reviewer’s detailed review.

Comment 6: Page 7/8, Limitations: Please clarify 'resulted in a high attrition rate'

Reply 6:

Yes, we thank for the reviewer’s insightful comment. We totally appreciate.

The sentence 'resulted in a high attrition rate' was originally meant to indicate that our article is a retrospective article. However, after consideration, we decide to delete this sentence in limitation section to avoid unnecessary confusion for readers on Page 7.

Comment 7: Correct few typos

Reply 7:

Yes, we thank for the reviewer’s comment. We totally appreciate.

We have corrected the typos as below:

“retrospectivegneal space” to “retropharyngeal space” as comment 5 displayed.

“retrophargeal space” to “retropharyngeal space” on Figure 2 legend, Page 6.

“retrophagyngeal space” to “retropharyngeal space” on Page 6 Line 16.